# Child and Adolescent Health Programs in Obesity and Depression: A Systematic Review and Meta-Analysis

**DOI:** 10.3390/nu17061088

**Published:** 2025-03-20

**Authors:** Ana Sarmiento-Riveros, María José Aguilar-Cordero, Juan A. Barahona-Barahona, Gabriel E. Galindo, Claudia Carvallo, Fernando A. Crespo, Héctor Burgos

**Affiliations:** 1Facultad de Salud, Escuela de Enfermería, Universidad Santo Tomás, Santiago 8370003, Chile; anasarmientori@santotomas.cl (A.S.-R.); jbarahona@santotomas.cl (J.A.B.-B.); 2CTS-367, Andalusian Plan for Research, Development and Innovation, University of Granada, 18071 Granada, Spain; mariajoseaguilar@ugr.es; 3Centro de Investigación e Innovación en Gerontología Aplicada CIGAP, Facultad de Salud, Universidad Santo Tomás, Santiago 8370003, Chileccarvallov@santotomas.cl (C.C.); 4Facultad de Economía y Negocios, Universidad Alberto Hurtado, Erasmo Escala 1835, Santiago 8340539, Chile; fcrespo@uahurtado.cl

**Keywords:** program, health programs, pediatric obesity, obesity, depression, child, adolescent

## Abstract

Obesity and depression are public health issues of increasing concern worldwide. This study aims to evaluate programs that address obesity and their impact on depressive symptoms in children and adolescents. Obesity and depression share a bidirectional relationship, where each can serve as both a cause and a consequence of the other. Methods: A systematic review and meta-analysis were conducted following PRISMA criteria, with the registration recorded under PROSPERO code (CRD42024550644). The selected publications report on intervention programs for obesity and depression in children and adolescents aged 6 to 18 years. The selection was from databases including PUBMED, SCOPUS, LILACS, COCHRANE, WOS, SciELO, and ScienceDirect, using PICOS criteria to define inclusion. ROB-2 and ROBINS-1 were applied to assess bias. Results: Out of 3376 articles reviewed, eight met the inclusion criteria, some including several programs. These programs varied in duration and type, demonstrating changes in reducing Body Mass Index (BMI) and depressive symptoms. However, evidence supporting the effectiveness of programs that address both conditions is limited, particularly in developing countries. Additionally, the results exhibit high heterogeneity due to the diversity of evaluation criteria and methodological approaches, highlighting considerable risks of bias. Conclusions: Intervention programs for obesity management show statistically significant effects on depressive symptoms, although there is heterogeneity in the designs for their standardization and long-term follow-up strategies; however, the evaluations consider DSM-5 and ICD-11 criteria, which contributes to homogeneity. It is vital to address these closely related issues from a multidimensional perspective, considering socio-emotional and psychological factors, and to promote early intervention to maximize effectiveness and enhance quality of life at various stages of development.

## 1. Introduction

Obesity and depression are diseases of high global prevalence that are incorporated as a priority in the context of public health [1]. The global prevalence of obesity has been projected to be 50% by 2035 in those over 5 years of age [2]. In Chile, in 2022, 26.2% of children under 15 years of age were obese [3]. Regarding mental health, around 20% of children and adolescents (NNA) have psychiatric disorders where the prevalence of depression in adolescents is comparable to that of adults, from 5% in early adolescence to 20% at the end of this period [4]. There has been an additional post-pandemic increase of 40.6% in Chile and 27.6% worldwide [5]. The development of this pathology is the result of various mechanisms, including diet and obesity [6,7]. Although these topics are often addressed separately, evidence suggests a link between obesity and mental health disorders, particularly the depressive symptoms.

Obesogenic behaviors generate an excessive accumulation of adipose tissue, constituting the state of obesity [8], triggering chronic inflammatory states, or humoral alterations, among others [9] present from preschool age [10]. Other complications include type 2 diabetes, dyslipidemia, hypertension, non-alcoholic fatty liver disease, obstructive apnea, and polycystic ovary syndrome [11]. In addition, psychosocial disorders such as bullying, social stigma, anxiety, and depression are reported [12]. Childhood obesity continues into adulthood, being associated with metabolic, cardiovascular, and physiological sequelae that can trigger depression or anxiety [13]. Obese children and adolescents have a high risk of major depression compared to healthy children [14]. Current self-image and body satisfaction are different in children and adolescents with and without obesity, impacting self-esteem and depressive symptoms that begin at school age [15]. Some data report short- and long-term mortality from 30 years of age [16]. Overweight/obesity in children and adolescents can lead to developmental problems, often influenced by stress that strongly impacts cognitive functions [17,18]. Several studies suggest that the lack of parental or guardian attention to children’s nutritional status may impact their early development [19,20].

For the diagnosis and monitoring of obesity, the Body Mass Index (BMI) is used, which relates weight (kg) and height (m^2^) [2]. The BMI-Z is used in children, classified by percentiles, where a result higher than 85 is overweight and higher than 95 is obese [21]. In other countries, they include electrical impedance [22]. Depressive states linked to obesity manifest themselves according to the age range [23,24]. The obesity–depression relationship is triggered by several mechanisms, where dietary factors are included, or the obesity state itself [6,7]. This condition can come from prenatal obesity that correlates with children and adolescents with overweight/obesity and parallel syndromes such as hypertension, and cardiovascular problems, among others [25]. Women are more likely to be diagnosed with depression when obesity is already present in the child population [26]. A high BMI correlates with depressive traits, which shows the relationship between these two states [27,28]. In addition, both pathologies have an inflammatory character, enhancing this associated pathophysiological phenotype [11]. In any case, evidence shows that the mechanisms that link depression with obesity include the immune and endocrine systems related to psychosocial aspects [29]. Although it is not conclusive whether obesity causes depression or whether depression leads to obesity, obese girls are more likely to develop depression compared to girls in a eutrophic state. At the same time, in boys, there is no significant difference [27,30].

Some studies of intervention programs in the child–youth population show effects of reducing body weight [1,31] however, they are scarce for depressive states [32]. Most of them highlight the risk of pathological typologies between cardiovascular diseases, bullying, and eating disorders, rather than the obesity/depression relationship, preventing the relational vision [33]. Interventions based on socio-ecological programs that modify child and family behaviors in educational contexts report greater effectiveness by integrating the primary health care system, schools, communities, and families [10,34]. Although interventions that increase physical activity influence the physical composition and biochemical markers in the child population, interventions that include diet, behavior, and family change combined with physical activity have more positive effects than physical activity alone [35,36,37].

Evidence suggests that obesity and depression are diseases that affect a large number of the population and are closely related [29,31], as they have multiple effects with consequences in the biological, psychological, and social spheres. Due to the scarce evidence of intervention programs that relate nutritional status with psycho-affective states in children and adolescents, especially treatments with a longer follow-up that can reduce combined neuroinflammatory states, the need arises to carry out this review. It aims to examine intervention programs that address the relationship between obesity and depression and their possible contribution to the improvement of depressive symptoms in children and adolescents that could influence adulthood.

## 2. Materials and Methods

The search, writing, and editing are based on the international statement of Preferred Reporting Items for Systematic Reviews and Meta-Analyses, PRISMA 2020 [38] and registered in the International Prospective Register of Systematic Reviews (PROSPERO: CRD42024550644 published on 2 August 2024).

### 2.1. Search Strategy

In terms of eligibility, the process begins with the identification and selection of original articles in English and Spanish in the electronic databases PUBMED, SCOPUS, LILACS, COCHRANE, WOS, SciELO, and ScienceDirect, between May and October 2024, to evaluate obesity programs that address depression. The search uses the PICOS model, which allows the orientation of the research question: Participants = children and adolescents from 6 to 18 years of age with obesity; Intervention = Health programs for obesity and depression; Comparison = control or contrast group; O = Weight or obesity parameter reduction, mood improvement, reduction in depressive symptoms; S = Randomized controlled trials (RCT) and non-randomized controlled trials (NECA) [39,40].

Keywords, previously verified in the DeCS and MeSH thesauri are as follows: “Program” OR “health programs” AND “Pediatric obesity” OR (Obesity*) AND “Obesity and depression” AND “Depression”, “Child*” OR “Childhood” and school-aged children, with Boolean AND/OR (See Appendix A).

### 2.2. Inclusion and Exclusion Criteria

The inclusion criteria according to the PICOS model were articles: (i) RCTs and NECAs, in English and Spanish, comparing interventions with programs that aim to reduce obesity and depressive symptoms (ii) in children and adolescents between 6 and 18 years of age, regardless of gender, socioeconomic status or race, and (iii) without a time limit. The exclusion criteria rule out all reviews or studies that (i) address only obesity or depression; (ii) do not present scales or assessment instruments; (iii) use animal models; (iv) hospitalized children and/or studies with children with any pre-existing health conditions other than obesity and depression; (v) and included adults.

### 2.3. Data Extraction

Three researchers consulted the databases using the methodology described independently for the study selection. Differences in opinion were discussed, reaching a consensus on the inclusion or exclusion of articles. The protocol for recording and coding the variables of interest and the results of the studies takes into account (a) the sample size of participants, (b) the country, (c) the age, (d) the type of program used, (e) the instruments used, and (f) the main results in terms of changes or improvement in obesity status and depressive symptoms. Specific data were extracted from the background of each study (author, year, country, and study design) and the description of the study participants according to intervention groups, total sample size and size of each group, characteristics of the intervention programs, and initial and final results for obesity and depressive symptoms according to the evaluation parameters of the instruments and scales used.

### 2.4. Risk of Bias and Quality Assessment

Three reviewers (J.B.B, H.B., and A.S.R) independently assessed the methodological quality of the studies. The risk assessment strategy considers the ROB-2 criteria for randomized controlled trials (RCT) in which the risks of bias (RS) derived from the processes involved are evaluated, including RS of randomization, RS of deviations from the intended interventions, RS of missing data, RS of results, RS in the measurement of the outcome, RS in the selection of the reported outcome. The articles are classified with a high, medium, or low risk of bias [41]. For non-randomized controlled studies (NRCT), the Risk of Bias in Non-Randomized Studies of Interventions (ROBINS-I) tool was used to identify possible sources of bias that could affect the validity of the results. It evaluates the RS of selection, the RS of intervention, the RS of confounding, the RS of measurement, the RS of reporting, and the RS of loss to follow-up. For each domain, the risk of bias is rated as low, moderate, serious, or unclear. This assessment is based on specific criteria that contemplate the quality and consistency of the information presented in the study [42].

### 2.5. Statistical Analysis

In the effect measures, this review analyzed data from scales measuring changes in obesity status and depressive symptom scores derived from psychological assessment tests based on DSM-5 and CIE-11 criteria. Secondary outcomes included age ranges, type of intervention, and program characteristics, such as duration, evaluated variables, and measurement instruments (See Table 1 and Table 2 in results).

For the meta-analysis, R software (version 4.4.2) was used. The dataset included complete pre-post reports, listed as mean, standard deviation, standard error, and sample size for the intervention and control groups across all RCTs. Values were included as they appeared in the original studies. Percentage values were adjusted relative to the total sample for standardization. Additionally, when data were provided as a standard error, they were converted to standard deviation by multiplying the standard error by the square root of the total sample size [43]. To estimate meta-analytic effects, a random-effects model was applied using the restricted maximum likelihood (REML) method, implemented in the *rma()* function from the *metafor* package in R. Using Cohen’s d to assess the effect size which quantifies the standardized mean difference between intervention and control groups. The confidence interval (CI) was set at 95% (*p* = 0.05) and study weights were determined based on the inverse variance of the estimated effect, computed using the *weights()* function from Metafor. The overall effect was considered statistically significant if the 95% CI did not include the null value of 0 in its range, as illustrated in the forest plot (*forestplot()* in R) (see Figures in results). Furthermore, heterogeneity and homogeneity statistics were calculated and are expressed in *I*^2^ and Q statistics. Given the variability among the studies, a random-effect model was implemented to account for potential bias. Publication bias was assessed through Egger’s regression test (*regtest()* in R) followed by the funnel plot (*funnel()* in R) allowing for the evaluation of asymmetry in order to study heterogeneity.

## 3. Results

### 3.1. Selection of Studies

A total of 3376 articles were examined from the electronic databases PUBMED, SCOPUS, LILACS, COCHRANE, WOS, ScienceDirect, and SciELO, plus 10 articles selected by other means. Twenty-two articles were eliminated for being duplicates. A total of 3032 were excluded by title and abstract according to inclusion/exclusion criteria. Likewise, after analysis of 322 full texts, 306 were eliminated. Of the remaining 16 studies, eight articles were excluded due to a lack of results for obesity and depression [44,45,46,47] or not being randomized controlled studies [48,49,50,51]. Finally, this review and meta-analysis include 8 reviewed articles [52,53,54,55,56,57,58,59], as shown in Figure 1.

### 3.2. Overall Characteristics of the Studies

This review examines the programs implemented for the treatment of obesity and depression in children and adolescents as a whole, regardless of the initial diagnosis of each pathology. Of the eight studies selected, only one [57] corresponds to a non-randomized controlled trial (NRCT). There is little evidence of programs involving this relationship as a whole. The ages in the studies range from 7 to 18 years with a mean of 15.6 years. With one exception, the studies are concentrated in high-income countries. The studies consider a sample population from 40 to 2400 participants. The types of programs include combinations of therapies, diet, and physical exercise, also behavioral therapy [52], interventions with diet and exercise [53,55], interventions with diet and behavioral therapy [47,48], or interventions with diet, exercise with behavioral therapy [58,59] (See Table 1).

**Table 1 nutrients-17-01088-t001:** Overall details of the selected studies.

Authors	Type of Study	Country	Age Range (Years)	Total Sample (N)	Type of Intervention
Barnes and Kristeller, 2016 [52]	RCT	USA	15–17	40	Behavioral Therapy
Goldfield et al., 2015 [53]	RCT	Canadá	14–18	304	Diet/Exercise
López et al., 2021 [58]	RCT	USA	14–18	76	Diet/Exercise Behavioral Therapy
Luca et al., 2014 [57]	NRCT	Canadá	14–17	116	Diet Behavioral Therapy
Sen et al., 2018 [59]	RCT	Turquía	9–12	108	Diet/Exercise Behavioral Therapy
Strugnell et al., 2024 [54]	RCT	Australia	13–16	2400	Diet/Exercise
Vidmar et al., 2022 [56]	RCT	USA	14–18	117	Diet Behavioral Therapy
Williams et al., 2019 [55]	RCT	USA	8–11	175	Diet/Exercise

NRCT = non-randomized controlled trials.

The programs present a variety of activities, sometimes more focused on obesity and other times more on mental health. The development time of the programs ranges between 6 and 24 months, with variations between 6 and 92 h of intervention. The BMI or z-BMI are frequently used for obesity assessments, and in mental health tests they use scales to evaluate depressive symptoms, which are all based on the criteria set out in the DSM-5 and CIE-11. In these latter mental health scales, they also considered other types of symptoms such as anxiety, anger management, and binge eating, among others, but they exceeded the objectives of this study (See Table 2).

**Table 2 nutrients-17-01088-t002:** Details of the programs.

Author/Year	Intervention Program	Extension (Months)	Partial Duration (Weeks)	Direct Intervention (Hours)	Variables Evaluated
Barnes and Kristeller, 2016 [52]	Program MB-EAT-A	6	12	9	BMI Diet Exercise Binge eating Feeding Depression Risk behaviors
Goldfield et al., 2015 [53]	Resistance, aerobic and combines traditional program	6	22	7–16.5	Mood Body image Physical self-perceptions Overall self-esteem
López et al., 2021 [58]	Management program with APP	6	24	12–24	BMI Executive function Childhood depression
Luca et al., 2014 [57]	Program STOMP	12	48	no informed	BMI quality of life Childhood depression Readiness to change HOMA-IR Diet Waist circumference Physical activity
Sen et al., 2018 [59]	Comparative program of family behavioral intervention vs. Kaledo game	3	no information	6	Psychiatric symptoms Physical activity Diet Anthropometry (weight, height, BMI, Z-BMI)
Strugnell et al., 2024 [54]	Program Healthy Together Victoria	24	12	no informed	BMI Waist circumference School Health Quality of diet Quality of life Depressive symptoms
Vidmar et al., 2022 [56]	mHealth intervention program with APP	6	24	16–18	BMI Food addiction Executive function Behavior Emotion Cognition Depression Stress
Williams et al., 2019 [55]	Exercise program	8	32	92	Quality of Life Anger Self-esteem Body composition Fat percentage Cardiovascular fitness.

### 3.3. Findings of the Systematic Review

(i)Effect of obesity programs

Studies with initial and final obesity data do not present relevant significant reductions [54,59]. However, it is noteworthy that the intervention at least stops the progression of obesity [54,55]. In the contrast or control groups, the indicators show an increase in the obesity condition [54,57]. Interestingly, Williams et al. (2019) show that men improve their waist circumference values, with a decrease in the consumption of sugary drinks compared to women, but without differences in the quality of life related to depressive symptoms [55].

(ii)Effect of programs for depressive symptoms

Regarding the reported pre- and post-evaluation depressive symptoms, all the selected studies use validated instruments that include the DSM-5 and CIE-11 criteria, which are comparable to the evaluation of depressive symptoms. In five of them [53,54,55,57,58], the values show that obesity intervention programs improve mental health in the child–youth population, where the greatest effect is related to programs that include resistance to physical activity. Also, of interest are the results of Luca et al. (2014), where they show that adolescents with greater depressive symptoms have a greater probability of executive dysfunction, a situation that complicates their overall health status [57,58].

A special mention is needed for those programs that include physical activity in which resistance training reduces depressive symptoms and the fat percentage [53]. Likewise, moderate or intense aerobic exercise improves eating habits in favor of low-calorie and low-fat foods [52] and a significant decrease in BMI [59]; however, they do not present a significant effect on depressive symptoms.

In summary, the intervention programs address obesity and psychological health, including depressive symptoms. The final results of psychological health after the intervention are shown, however, as noted, it is incomplete for the nutritional status of the participants. The programs suggest that including physical exercise presents significant changes in mental health (See Table 3).

### 3.4. Risk of Bias in Individual Studies

The risk of bias (SR) for RCTs [52,56,58,59] was assessed using the Cochrane ROB-2 tool [41]. Studies were classified as either high, unclear, or low risk of bias. One study had a high SR for the randomization process, while three had an unclear risk of bias as they lacked specificity in the process, and four were classified as low SR as the methods and description were adequate. In the SR for deviations from planned interventions, one study had a high SR for allocation concealment, two studies had some unclear risk as they did not specify the methods used, and five had a low risk for using adequate methods. Two studies were assessed with a high SR for missing data on outcomes due to high dropout rates, and six studies were assessed with a low SR associated with incomplete data on outcomes. In the measurement of the results, three studies had a high SR, one with some SR and four studies with a low SR. Finally, regarding the SR in the selection of the reported result, three studies with high SR, four with some SR, and one with low SR were evaluated (Figure 2).

The SR assessment for NRCT was performed using the ROBINS-I tool. In the assessment outcome, domains 1, 4, and 6 presented low SR. However, in domain 2, there are some concerns about the measurement of outcomes; in domain 3, there are some concerns about controlling for factors in the intervention-outcome relationship; in domain 5, there are concerns about how participants were selected for this study; and in domain 7, the concerns are in the way outcomes are reported or in the choice of which outcomes are reported [42]. In summary, among the studies analyzed, six were classified as having high risk of bias [52,53,55,58,59] and three as a having risk of bias with some concerns [54,56,57]. The summary of findings and risk of bias among contributing studies is presented in Table 4.

### 3.5. Meta-Analysis

The effects of the programs were confirmed in both experimental and control groups, where the overall effect with statistical significance (Cohen’s d = −1.465, *p* = 0.016) and confidence intervals indicate a reduction in depressive symptoms associated with the intervention programs (See Table 5), according to Harrer et al., (2022) [60].

Therefore, programs in people with obesity reduce the effects of depressive symptoms because each confidence interval does not exceed the critic limit, so the programs show this statistically significant trend (See Figure 3).

The random effects model is robust considering the variation between the real differences in the individual studies. τ^2^ = 4.69 means extreme heterogeneity (99.7%) and congruent homogeneity (Q = 2124.99; *p* = 0.00), so the individual effect sizes are very diverse. These values could show patterns related to implicit moderating factors, such as age, gender, type of intervention, and methods already analyzed in Table 4 by the risk of bias analysis. The variability between the effect sizes can be attributed entirely to chance rather than to real differences between the studies.

This is reflected in the evaluation of publication bias in the funnel plot, the Egger intercept beta coefficient (1.405; *p* = 0.026), reflecting the asymmetry observed in Figure 4 and Table 6, as well as the coefficient associated with the standard error (−11.891; *p* < 0.001).

The aforementioned shows the influence of the effect size of Goldfield resist [53] and Luca stomb [57] may present an impact given by the N considered. Globaly, the studies show statistical significance for most outcomes, except Goldfield aerobic [53] and Williams control [55] which cross the critical point. The López alone and control studies also reduce depressive symptoms, as well as others show that the program does not influence this condition (Goldfield combination [53], Williams exercise [55], and Strugnell intervention [54]). Finally, the overall effect of the programs shows a statistically significant decrease in depressive symptoms, strengthening the validity of the findings of this meta-analysis. Although heterogeneity is a frequent characteristic of clinical variability, it does not detract from this global result reflected in the axis marked in green, including the confidence interval.

This meta-analysis also shows the results of the contrast group presented in the studies. When analyzing the effects of the program with only the experimental groups, excluding the control group or the contrast group, the data from the forest and funnel plots present almost the same indicators of heterogeneity and homogeneity as those analyzed previously and also show the statistical significance regarding the reduction in depressive symptoms (See Figure 5).

The effect size is significant, and the forest plot also shows that depressive symptoms decrease due to obesity treatment. However, the data remains dispersed (See Table 7).

The funnel plot shows the results associated with each study, including the mean and standard deviation declared by the study. The peaks show the heterogeneity of the studies. However, most of the experimental results show that the symptoms decrease with the effect of the treatments on obesity, located on the left side. Precisely, those results of contrast or control groups are located on the right side of the graph. The blue part of the graph shows the average of the differences with a standard deviation added from the standard deviations of all the studies, assuming normal deviations. In this, some individual studies touch this limit. The average of the differences using the pooled standard deviation or tau value corresponds to the confidence interval with the unified standard deviation. In this, the studies are far from this limit, which is a sign of heterogeneity. The graph also shows the confidence interval at the widest point of the funnel because it covers more standard deviation. Note that the vertical axis goes between 0 and 1, because the highest significance level is at *p* = 0, and as it increases, it decreases, and that is why the cone becomes narrower. It is expected, which is why the peak does not mean anything conceptually, it only indicates the mean of the difference on the x axis. This answers the questions of why the inverse significance = 1/confidence level appears (See Figure 6).

The funnel plot and Egger’s Regression-Based Test show the heterogeneity of the studies but maintain the effect of the treatment on depressive symptoms (See Figure 6 and Table 8).

## 4. Discussion

This systematic review and meta-analysis show a limited amount of robust evidence derived from randomized controlled trials (RCTs) or non-randomized controlled trials (NRCTs) concerning interventions aimed at child and adolescent health, specifically addressing the relationship between obesity and depressive symptoms. Most existing studies tend to focus on the conditions of either obesity or depression, often providing treatment recommendations without sufficiently clear insights into their effects on this interrelationship [61]. Interventions primarily concentrate on weight reduction through dietary modifications, physical activity, and behavioral therapy, which may generate indirect effects on participants’ mental health, including depressive symptoms, as referenced in various other meta-analyses related to this association [13]. Noteworthy improvements in mental health outcomes have been observed in programs that incorporate anaerobic physical exercise using free weights or weight machines [52]. A meta-analysis has indicated that exercise significantly contributes to the reduction in depressive symptoms in children and adolescents, with the most prominent benefits associated with aerobic exercises performed for 40 to 50 min, three times weekly over a span of 12 weeks [14]. However, conflicting evidence exists regarding this relationship, as certain studies suggest that symptoms—especially those of a more severe nature—may not become apparent until later stages of adulthood, underscoring a subtle, long-term impact [62]. Emerging research suggests that neurobiological mechanisms underlie the obesity–depression relationship, orientated to cortico-frontal and limbic system dysregulation might be potential therapeutic targets for future interventions [63,64].

Studies indicate that the effects can be enhanced by combining intervention strategies such as behavioral or family therapy, along with diet and exercise, which may lead to a decrease in BMI and skinfold thickness [65]. However, this is not conclusive in reducing depressive symptoms [58,59]. While some studies suggest that interventions can enhance certain health aspects, such as cardio-metabolic parameters and executive functioning, significant reductions in BMI or depressive symptoms are not consistently realized [57]. This suggests that the relationship between obesity and mental health is complex; therefore, changes in diet and physical activity do not necessarily guarantee improvement. They may need cognitive factors like executive functioning, which influences cognitive inhibition, planning, and working memory in our behavior, especially regarding those that affect stress [66,67]. In this respect, decreased cognitive inhibition, working memory, and metacognitive components are common in children with obesity, impacting cognitive functioning and leading to reduced cortical thickness in the prefrontal cortex [68], which is related to unhealthy eating behaviors [60]. Notably, this condition has shown factors that contribute to inflammatory states in the brain, typical of depressive states, which may be the cause of cognitive decline and require further investigation in future studies [69].

It is notable that most studies addressing intervention programs for obesity and depression in databases are from countries with high or medium development levels, with few reports from Latin American countries. In these regions, only the high rates of childhood obesity and depression are reported, but not effective programs or interventions [70,71,72,73]. Among the evidence found in the area, one review stands out, considering publications from three Central and South American countries about the relationship between obesity and depression. It focuses on the influence of parents with depressive symptoms on children73, with obesity, leaving aside genetic and epigenetic hypotheses that may also affect this relationship [74]. Marco et al. (2020) [74] highlighted the shared causes of these two pathologies, where the mechanisms may overlap [25,26,75]. Some inflammatory markers, such as C-reactive protein and interleukin-6, have been linked to deregulated neurohormonal circuits in both obesity and depression [11]. In addition, the impact of adipokines and lipokines, such as leptin, adiponectin, and interleukins, among other neurochemicals, has been demonstrated in the alteration of the pathophysiological mechanisms involved in both conditions, affecting neuroplasticity and regulatory circuits in the stress axis, as well as other neuroimmunoinflammatory responses, even addressing aspects related to ventral vagal functioning [76,77]. Regarding dietary factors, there is a wealth of information on the effect of the microbiota that triggers regulatory elements capable of preventing inflammatory phenomena linked to obesity and depression [78]. Future interventions should focus on targeting the shared obesity–depression condition through inflammatory pathways, particularly those affecting executive function, a key cognitive process implicated in both conditions [79].

The age ranges considered in the studies primarily pertain to adolescents, with limited information available regarding early childhood or pre-adolescence. Among adolescents, the impact of stress, anxiety, and depression on the development of obesity is significant. This is where increased emotional eating and decreased physical activity intersect, highlighting a critical need for early intervention and preventative measures [64]. One possible reason for this can be traced back to the characteristics of the psycho-evolutionary period. Research indicates that the connection between self-esteem and depressive symptoms begins during the school years, with certain variables related to depression, such as self-image, becoming more pronounced [15]. This underscores the importance of recognizing earlier symptoms or prodromes that may influence this condition. Additionally, the nature of the programs designed to address either the conditions or the combined issue of obesity and depression requires consideration. For instance, few programs meet a standardized approach that allows for meaningful comparisons, as BMI or z-BMI data are primarily utilized for obesity while various instruments are deployed for measuring depressive symptoms. The studies conducted by Williams et al. (2019) [55] and Goldfield et al. (2015) [53] share a common theme: they emphasize the importance of physical activity in enhancing quality of life and alleviating depressive symptoms among overweight and obese adolescents. Williams et al. (2019) [55] found that both an exercise program and a sedentary program led to improvements in self-esteem and a reduction in depressive symptoms, though neither affected anger management. This suggests that participation in a program, whether active or passive, can positively influence psychological well-being. Conversely, Goldfield et al. (2015) [53] demonstrated that resistance training specifically alleviated depressive symptoms, indicating that different forms of activity can exert varying effects on adolescents’ mental health.

Few studies provide longitudinal monitoring of the obesity–depression comorbidity. The reviewed interventions exhibit high variability in terms of duration, frequency, and assessment instruments, limiting the generalizability of findings. A standardized methodology for future trials is necessary to improve comparability across studies. An interesting meta-analysis focused on adults with 25 studies shows that the reduction in caloric intake given by carbohydrates decreases depressive symptoms, although, like our study, with biases and incomplete data in the presentation of the results [73]. In the child population, the obesity/depression relationship is linked to other causes; one of them is the nutritional status of mothers and fathers, which influence depressive symptoms, in addition to the development of personality characteristics related to isolation, and self-esteem, among other aspects evaluated, but not resulting from the execution of a program [80]. On the other hand, it is surprising that most publications report incomplete data on both obesity [47,53,56,58] and depression [52,56,59]. This is evidenced by the bias analysis where the findings presented in the articles may not be conclusive enough to generalize this obesity/depression relationship (See Table 3 and Figure 5). Dietary interventions to reduce depressive symptoms in the child–youth population present some favorable results, however, the weaknesses in the structure and methodological design used and the presentation of the results also leave this conclusion pending [81]. On the other hand, the high dropout rate shown by some studies in behavioral and group or family interventions relativizes the effectiveness of the interventions in reducing obesity indicators, with weakness also in the impact on depressive symptoms, an aspect to be taken into account in future programs that can be implemented that take into account the characteristics of children and adolescents, especially those related to depressive symptoms [59].

In summary, the strengths of this meta-analysis are that it addresses a topic that has been studied little regarding the effect of programs that address obesity and depressive symptoms. Future research should prioritize longitudinal studies that examine early-life interventions using neuroimaging biomarkers to assess their long-term impact on both metabolic and psychological health [82]. Changes in depressive symptoms including obesity treatments is a strategy that can be strengthened, probably due to the similar physiological and neuropsychological changes that underlie the cause. The implications for mental health may be promising because most pathologies are closely related, as in this specific case of obesity/depression.

Given the limitations of this study, the high heterogeneity of the results is a point to address, although the data presented by the selected articles are related to symptoms presented in the diagnostic manual for mental illnesses, DSM-5 or CIE-11 for depressive symptoms, which could contribute to their homogenization. This is not explicitly reflected in the analysis, nor is the high clinical variability that these types of pathologies present, despite the data being evaluated quantitatively.

## 5. Conclusions

Obesity programs show a statistically significant effect in reducing depressive symptoms in children and adolescents. Nevertheless, the instruments and methodologies used are diverse, but all follow international standard criteria to evaluate this effect on such depressive symptoms. The magnitude of the effect is consistent, despite the high heterogeneity and evidence of publication bias, which is why the random effects model was used in the meta-analysis. In obesity, the effect is not significant. Future studies could focus on strategies that integrate the physical and mental health of this segment of the child–youth population that encompasses various population and sociodemographic contexts, in a multisectoral approach. This would intend to link psychological and emotional factors in the design of programs to ensure long-lasting effectiveness and improve the quality of life in this age range.

## Figures and Tables

**Figure 1 nutrients-17-01088-f001:**
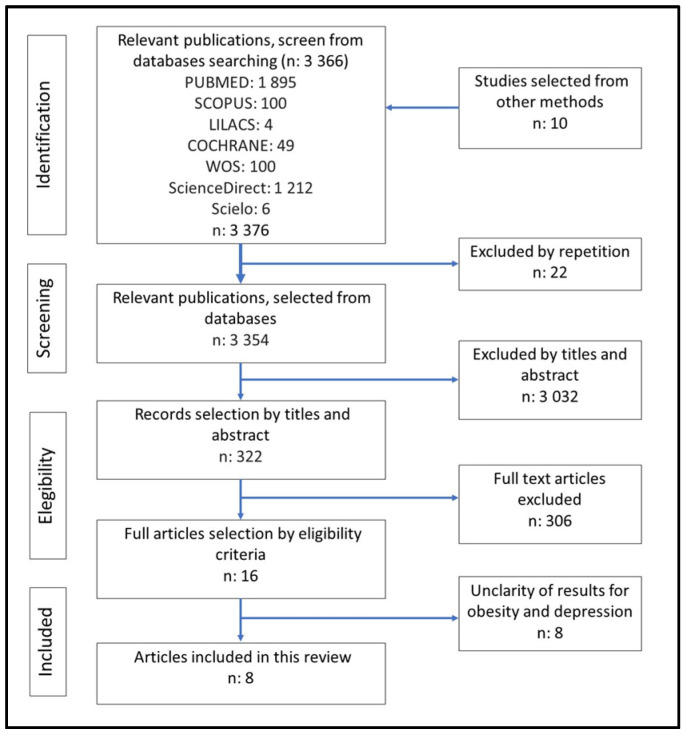
Flowchart for the selection of articles regarding programs for obesity and depression in children and adolescents, according to the PRISMA protocol.

**Figure 2 nutrients-17-01088-f002:**
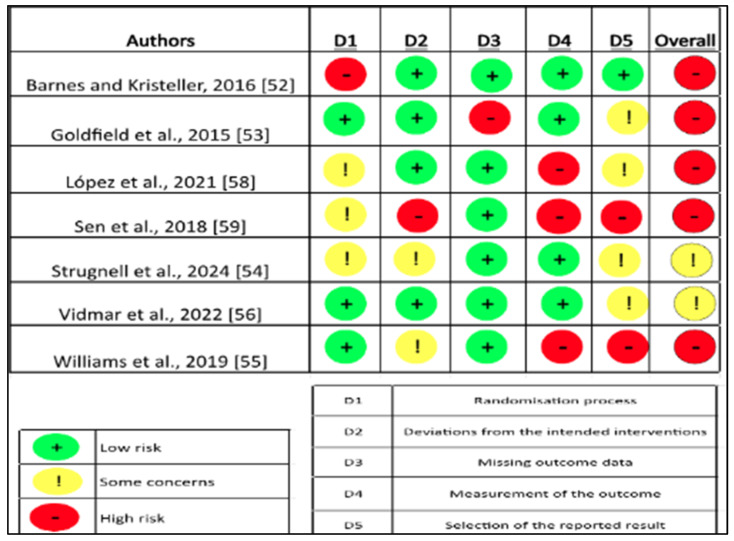
Risk of bias assessment for RCTs articles [52,53,54,55,56,58,59] respectively using Rob-2.

**Figure 3 nutrients-17-01088-f003:**
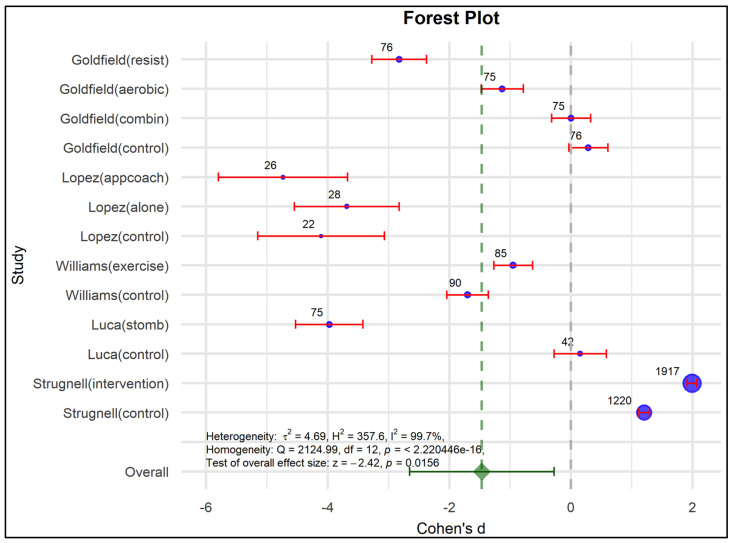
Forest plot to examine effect size and data dispersion in the publications in both experimental and control groups.

**Figure 4 nutrients-17-01088-f004:**
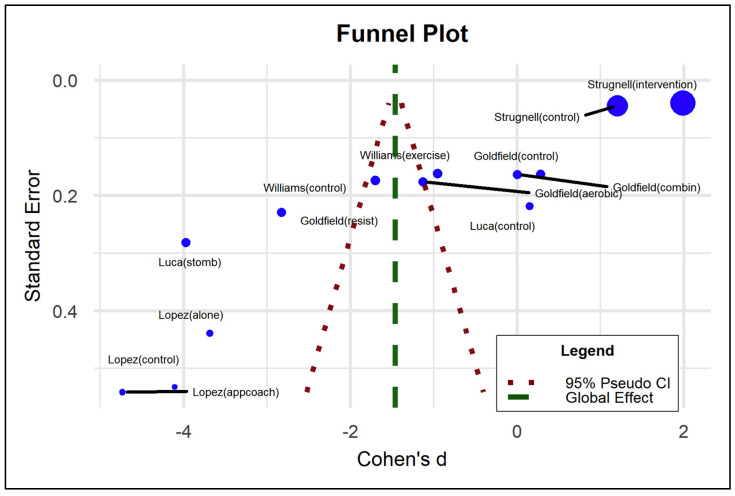
Funnel plot that examines heterogeneity of the results in the data presents in the articles.

**Figure 5 nutrients-17-01088-f005:**
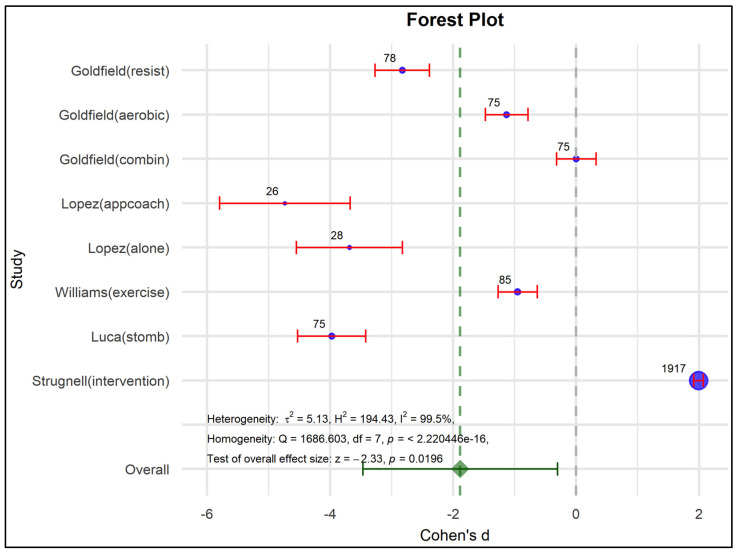
Forest plot with only changes in depressive symptomatology in obesity treatment in experimental groups.

**Figure 6 nutrients-17-01088-f006:**
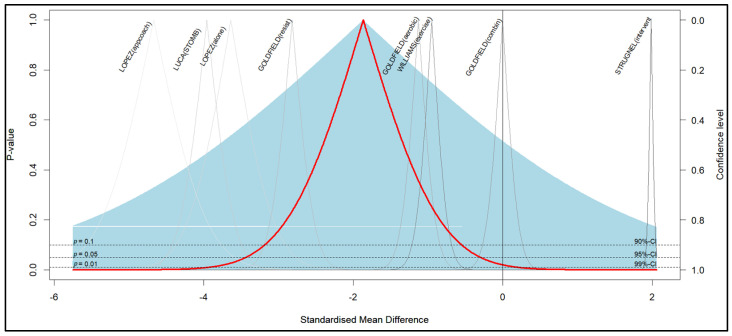
Funnel plot that examines heterogeneity of the results in the experimental groups.

**Table 3 nutrients-17-01088-t003:** Results by program regarding obesity and depressive symptoms.

Author/Year	Program (Interventions)	N	Obesity			Depression		
			Measuring Instrument	Initial	Post Intervention	Measuring Instrument	Initial	Post Intervention
Goldfield et al., 2015 [53]	(a) Aerobic training	75	Z-BMI	34.6	no informed	* BRUMS	2.5 (0.3)	2.1 (0.4)
	(b) Resistance training	78		35.1	no informed		2.7 (0.3)	1.7 (0.4)
	(c) Combined training (Aerobic + Resistance)	75		34.6	no informed		2.6 (0.3)	2.6 (0.4)
	(d) Control without exercise	78		34.1	no informed		2.8 (0.3)	2.7 (0.4)
López et al., 2021 [58]	(a) App Coach	26	Z-BMI	no informed	no informed	* CES-DC	8.38	5.22
	(b) App Alone	28		no informed	no informed		8.21	5.26
	(c) Control	22		no informed	no informed		7.91	5.41
Strugnell et al., 2024 [54]	(a) Intervention	1917	Z-IMC	34.4	33.5	* SMFQ	4.5	5.4
			Waist circumference	75.7	75.2			
			Abdominal obesity	16.5	17.8			
	(b) Comparison	1220	Z-BMI	29.6	32.5		5.1	5.7
			Waist circumference	74.4	75.2			
			Abdominal obesity	13	16			
Luca et al., 2014 [57]	(a) STOMP Intervention	75	BMI	44.8	0.08 ± 0.3	* CDI	11.9 ± 4.2	(11.9 ± 4.2)–(3.6 ± 1.4)
	(b) Comparison	42		34.5	0.7 ± 0.2		6.0 ± 9.0	(6.0 ± 9.0)–(0.09 ± 1.0)
Barnes and Kristeller, 2016 [41]	(a) MB-EAT-A	18	BMI	32.9 ± 8.8	33.0 ± 9.4	* BASC	no informed	(ρ = −0.008, *p* = 1.0)
	(b) Health education control	22		32.0 ± 9.4	32.0 ± 9.4		no informed	no informed
Williams et al., 2019 [55]	(a) Aerobic exercise	85	Body fat	38.3 ± 6.9	38.3 ± 6.9	* CDI	(7.6 ± 6.6)	(6.3 ± 5.2)
	(b) Sedentary attention control	90		36.7 ± 7.3	36.7 ± 7.3		(8.1 ± 7.5)	(6.8 ± 5.9)
Vidmar, et al., 2022 CES-DC [56]	(a) control	39	%BMIp95c	125.43 [101.56, 197.93]	no informed	* CES-DC	8	no informed
	(b) APP	39		129.53 [104.61, 193.47]	no informed		7	no informed
	(c) APP coach	39		129.53 [104.61, 193.47]	no informed		8.5	no informed
Sen et al., 2018 [59]	(a) Behavioral	12	BMI	25.36 ± 2.37	24.43 ± 2.33	* CDI	no informed	6.30 ± 5.66
	(b) Game	12		26.81 ± 3.10	26.24 ± 2.67		no informed	8.92 ± 4.50

* See acronyms in Appendix B.

**Table 4 nutrients-17-01088-t004:** SR summary and conclusions.

Author/Year	Type of Study	Conclusions	Assessment Instrument	Risk of Bias
Barnes Kristeller, 2016 [52]	RCT	The MB-EAT-A program can improve the dietary habits of school-aged adolescents. Feeding programs are a means to address obesity in high-risk youth.	ROB-2	High risk
Goldfield et al., 2015 [53]	RCT	Resistance training, alone or in combination with aerobic training, can provide psychological benefits in overweight or obese adolescents.	ROB-2	High risk
López et al., 2021 [58]	RCT	Family participation in intervention programs is related to greater program attendance.	ROB-2	High risk
Luca et al., 2014 [57]	NRCT	The STOMP program did not show a significant reduction in BMI, but there were improvements in cardiometabolic, psychological, and behavioral outcomes.	ROBINS-I	Moderate risk
Strugnell et al., 2024 [54]	RCT	The program produces improvement in waist circumference and in the consumption of sugary drinks per day. For girls, there were no statistically significant differences.	ROB-2	Moderate risk
Vidmar et al., 2022 [56]	RCT	No significant changes in nutrition parameters were observed in the intervention promama, but they were positive for depression and stress.	ROB-2	Moderate risk
Williams et al., 2019 [55]	RCT	Sedentary programs that include games and activities of interaction with adults and peers, as well as a behavioral structure, may be more beneficial for mood than those focused solely on physical exercise.	ROB-2	High risk
Sen et al., 2018 [59]	RCT	Family-based behavioral group intervention and play-based intervention (Kaledo) were equally beneficial in lowering childhood BMI. So they can be used in the treatment of childhood obesity.	ROB-2	High risk

**Table 5 nutrients-17-01088-t005:** Effect Size Estimates in both experimental and control groups.

	Effect Size	Std. Error	Z	Sig. (2-Tailed)	95% Confidence Interval	
					Lower	Upper
Overall effect	−1.465	0.6057	−2.418	0.016	−2.652	−0.277

**Table 6 nutrients-17-01088-t006:** Egger’s regression-based test for meta-analysis of the forest plot, shown in Figure 3.

Parameter	Coefficient	Std. Error	t	Sig. (2-Tailed)	95% Confidence Interval	
					Lower	Upper
(Intercept)	1.405	0.5458	2.574	0.026	0.203	2.606
SEb	−11.891	19.550	−6.082	<0.001	−16.194	−7.588

Random-effects meta-regression, Standard error of effect size.

**Table 7 nutrients-17-01088-t007:** Effect Size Estimates in experimental groups.

	Effect Size	Std. Error	Z	Sig. (2-Tailed)	95% Confidence Interval	
					Lower	Upper
Overall effect	−1.885	0.8077	−2.333	0.020	−3.468	−0.302

**Table 8 nutrients-17-01088-t008:** Egger’s Regression-Based Test for meta-analysis of the forest plot, shown in Figure 5.

Parameter	Coefficient	Std. Error	t	Sig. (2-Tailed)	95% Confidence Interval	
					Lower	Upper
(Intercept)	1.292	0.7634	1.692	0.142	−0.576	3.159
SEb	−12.589	26.536	−4.744	0.003	−19.082	−6.096

Random-effects meta-regression, Standard error of effect size.

## Data Availability

No new data were created or analyzed in this study.

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
