# Peer review of "Child and Adolescent Health Programs in Obesity and Depression: A Systematic Review and Meta-Analysis"

_nutrients, 2025, doi:10.3390/nu17061088_

Round 1
Reviewer 1 Report
Comments and Suggestions for Authors
Comments to authors
The authors conducted a systematic review and meta-analysis of programmes aimed at young people and their effects on obesity and depression. Although interesting, the authors should restructure the introduction, methodology and results to improve reader understanding. In addition, and more importantly, they should provide a more detailed description of the statistical analyses carried out and provide appropriate references.
Abstract
· The PROSPERO code goes together, not separately, i.e. CRDXXXXXXX, not CRD XXXXXXX.
· A time filter is not justified. It should be searched from the beginning to the present.
· "only 8 met...": Avoid value judgements. The results should indicate that 8 were included, without "only".
· When dealing with trials, it is more correct to use the term effect or efficacy rather than effectiveness.
Introduction
· I suggest that authors reduce the length of this section. It should be no more than 1 page in total and ideally consist of three paragraphs: first paragraph, describing the problem of obesity and depression in the study population; second paragraph, the association between obesity and depression; third paragraph, the available interventions, with particular emphasis on the ones you are studying; fourth paragraph, why you are doing your study and what the aim is. And the paragraphs should be no longer than 12-15 lines. The problem with such long introductions is that they make it difficult for the reader to interpret the data.
Methods
An unstructured methodology makes interpretation difficult. The methodology should be structured in the following sections:
- Search strategy
- Inclusion/exclusion criteria
- Data extraction
- Risk of bias assessment
- Quality of evidence assessment
- Statistical analysis
Although the above is purely structural, the section on statistical analysis should also be expanded. This section should describe how the meta-analyses were conducted (fixed effects? Random effects? Categorisation of heterogeneity? P-value for heterogeneity? Assessment of publication bias? Meta-regressions? What software was used?)
Results
· In systematic reviews, the first step is usually to eliminate duplicates and then to read the title and abstract.
· "Finally, this review and 165 meta-analyses include 8 articles reviewed below": includes the citations of the included articles.
· "Specific characteristics of the studies": I think the authors were referring to "Findings of the systematic review" with this section. It is good that the authors include the results of the systematic review (not just the meta-analysis). Sometimes researchers just do the meta-analysis without doing a synthesis of the systematic review results. However, this section should be reduced to two paragraphs, one for the effect on obesity and one for the effect on depressive symptoms. And focus only on the results, more synthesised, rather than explaining the design and aim of each study (in reality, the design and so on should be in a table). Specifically, the authors report this in Tables 3 and 4).
· The section where the authors describe the meta-analysis and other derived analyses should have its own subsection called "Meta-analysis".
· The authors assessed heterogeneity, publication bias, meta-regressions, etc., but did not include them in the methodology. They should include it and cite it, and report only the results in the results section.
· "Figure 3. Forest plot to examine heterogeneity and effect size of publications: First effect size and then heterogeneity, not the other way round.
Discussion
· The first paragraph should summarise the results of the systematic review and meta-analysis.
· Although some limitations are mentioned throughout the discussion, a specific limitations paragraph should precede the conclusions. The acknowledgement of limitations is a strength of the study.
· The conclusions should be summarised in about 8 lines and answer the aims of your study.
Author Response
Dear Reviewer 1.
First of all, we sincerely appreciate your valuable comments, which have helped us enhance the clarity and precision of our study.
Below, we present our responses to your suggestions one by one in a table (see attached).
We look forward to hearing from you regarding this matter.
Best regards
The Authors

Reviewer 2 Report
Comments and Suggestions for Authors
The article “Child and Adolescent Health Programs in Obesity and Depression: a Systematic Review and Meta-analysis” aims to evaluate programs targeting obesity reduction and depressive symptoms in children and adolescents. Despite employing PRISMA guidelines and being registered in PROSPERO, the article demonstrates several limitations that may affect its scientific value and potential for publication.
Firstly, the number of included studies (only 8 out of 3366 initially reviewed) and their high heterogeneity (I² = 100%) limit the generalizability of the findings. The lack of standardization in assessment tools for both obesity (e.g., varying approaches to BMI and Z-BMI) and depression (multiple scales, lack of comparability) significantly hinders interpretation. Moreover, most studies are derived from high-income countries, excluding crucial data from a global context, especially from developing nations.
Another significant limitation is the high level of systematic bias risk (ROB-2 and ROBINS-I), pointing to deficiencies in randomization, lack of transparency in reporting results, and high participant dropout rates in some studies (e.g., 74% in one study). This suggests that the findings may be compromised by methodological errors and insufficient control of confounding variables.
Figures 1, 4, 5, and 6 require improvements in graphic quality and readability. In particular, the funnel plot (Figure 4) appears incomplete in describing asymmetry, and the PRISMA diagram (Figure 1) includes excessive information, making it difficult to interpret.
To improve the quality of the work and enable its publication, the authors should:
- Provide a more detailed description of statistical analysis methods to enhance the clarity of result interpretation.
- Fill in missing data on baseline and final outcomes in all studies, ensuring greater precision in reporting results.
- Apply uniform assessment tools for both conditions (obesity and depression) to increase the reliability and comparability of the results.
- Present a detailed plan to control confounding variables and minimize systematic bias.
- Enhance the quality of the figures, especially their labels and annotations, to make them clearer and more comprehensible for the reader.
If the above improvements are not implemented, the article should be deemed as requiring major revisions before acceptance. In its current form, despite utilizing recognized methodological frameworks, it does not meet the quality standards expected of scientific publications, especially those based on evidence-based medicine (EBM).
Author Response
Dear Reviewer 1.
First of all, we sincerely appreciate your valuable comments, which have helped us enhance the clarity and precision of our study.
Below, we present our responses to your suggestions one by one in a table. See attached.
We look forward to hearing from you regarding this matter.
Best regards
The Authors

Round 2
Reviewer 1 Report
Comments and Suggestions for Authors
Comments to authors
The authors have addressed most of the issues, except for a few that I detail below.
- “Publica-18 tions reporting intervention programs for obesity and depression in children and adoles-19 cents aged 6 to 18 years, published from 2012 onwards”: The authors continue to maintain the time limit in the abstract. Delete.
- “only eight met the inclusion criteria”: delete only.
- “this review and meta-analysis include 8 reviewed articles [52 , 59],”: change [52 , 59] by [52-59]
- It would be appreciated if the resolution of the figures could be increased.
Author Response
Thank you for your comments and below are the suggested changes
Best regards
Comments 1: “Publications reporting intervention programs for obesity and depression in children and adolescents aged 6 to 18 years, published from 2012 onwards”: The authors maintain the time limit in the abstract. Delete.
Response 1: Thank you for pointing this out. We agree with this comment. Therefore, we have edited the text highlighted in red in the attached document. See page 1, lines 18 – 20. This is consistent with what appears in the methodology.
Comments 2: “Only eight met the inclusion criteria”: delete only.
Response 1: Thank you for pointing this out. We agree with this comment. Therefore, we have removed “only” from line 23 on page 1. See the attached document. We have removed the word “only” on line 23 of page 1, this is consistent with line 191 on page 5 in the results section. See the attached document.
Comments 3: “This review and meta-analysis include 8 reviewed articles [52, 59],”: change [52 , 59] by [52-59]
Response 1: Thank you for pointing this out. We agree with this comment. Therefore, the change [52-59] was made on line 193, page 5. See the attached document.
Comments 4: It would be appreciated if the resolution of the figures could be increased.
Response 4: Thank you for the comment, we agree. Consequently, the quality of figure 1 (lines 196-198, p. 5); Figure 2 (lines 273,274 p. 9); Figure 3 (lines 295 - 298, p. 10); Figure 4 (Lines 316, 317, p. 11); Figure 5 (Lines 327, 328, p. 12)and Figure 6 (line 354, 3355, p. 13) has been improved.
Thanks again

Reviewer 2 Report
Comments and Suggestions for Authors
The changes made are satisfactory.
Author Response
Dear Reviewer.
Single comment: The modifications implemented are satisfactory.
Response: We appreciate your suggestions and feedback that improved the manuscript.
Best regards,